# Neutralizing Ability of a Single Domain VNAR Antibody: In Vitro Neutralization of SARS-CoV-2 Variants of Concern

**DOI:** 10.3390/ijms232012267

**Published:** 2022-10-14

**Authors:** Blanca J. Valdovino-Navarro, Salvador Dueñas, G. Isaí Flores-Acosta, Jahaziel Gasperin-Bulbarela, Johanna Bernaldez-Sarabia, Olivia Cabanillas-Bernal, Karla E. Cervantes-Luevano, Alexei F. Licea-Navarro

**Affiliations:** Biomedical Innovation Department, Centro de Investigación Científica y Educación Superior de Ensenada, (CICESE), Ensenada 22860, Baja California, Mexico

**Keywords:** COVID-19 neutralizing antibody, SARS-CoV-2, single-domain antibody, spike protein, VNAR therapy

## Abstract

Severe Acute Respiratory Syndrome Coronavirus 2 is the causal pathogen of coronavirus disease 2019 (COVID-19). The emergence of new variants with different mutational patterns has limited the therapeutic options available and complicated the development of effective neutralizing antibodies targeting the spike (S) protein. Variable New Antigen Receptors (VNARs) constitute a neutralizing antibody technology that has been introduced into the list of possible therapeutic options against SARS-CoV-2. The unique qualities of VNARs, such as high affinities for target molecules, capacity for paratope reformatting, and relatively high stability, make them attractive molecules to counteract the emerging SARS-CoV-2 variants. In this study, we characterized a VNAR antibody (SP240) that was isolated from a synthetic phage library of VNAR domains. In the phage display, a plasma with high antibody titers against SARS-CoV-2 was used to selectively displace the VNAR antibodies bound to the antigen SARS-CoV-2 receptor binding domain (RBD). In silico data suggested that the SP240 binding epitopes are located within the ACE2 binding interface. The neutralizing ability of SP240 was tested against live Delta and Omicron SARS-CoV-2 variants and was found to clear the infection of both variants in the lung cell line A549-ACE2-TMPRSS2. This study highlights the potential of VNARs to act as neutralizing antibodies against emerging SARS-CoV-2 variants.

## 1. Introduction

Severe Acute Respiratory Syndrome Coronavirus 2 (SARS-CoV-2) is a positive-sense RNA virus of the coronavirus family that is the causative agent of coronavirus disease 2019 (COVID-19) [1]. Since the first record of SARS-CoV-2 in late 2019, high transmission of the virus has led to a global pandemic that has resulted in a devastating global health and economic crisis. The spike (S) protein of SARS-CoV-2, which forms a trimer anchored to the viral membrane, is the major surface glycoprotein and mediates viral entry through cell receptor ACE2 [2]. The S protein makes use of its receptor binding domain (RBD) to engage ACE2, which is located on the surface of the host cell. The RBD-ACE2 interaction triggers a conformational rearrangement in the S protein, which is followed by the fusion of the viral and host-cell membranes [3,4]. Given that it is superficially exposed and essential for virus infectivity, the S protein is the main target of the human immune system to produce neutralizing antibodies against SARS-CoV-2 [5,6,7]. Additionally, therapeutic strategies based on monoclonal antibodies (mAbs) are directed at the spike RBD [8,9].

SARS-CoV-2 has undergone substantial evolution since it was first detected, which has resulted in the emergence of novel variants that harbor higher numbers of mutations than the previously identified variants. The World Health Organization (WHO) has classified the SARS-CoV-2 lineage as variants of concern (VOC). These variants have been associated with an increase in the transmissibility of SARS-CoV-2 and a reduction in susceptibility to the currently approved antibody and convalescent plasma therapies [10,11,12,13]. This represents a threat to the global population, as these variants continue to spread worldwide. The VOCs Delta (B.1.167.2), which emerged in October 2020 in India, and Omicron (B.1.1.529), which emerged in multiple countries in late 2021 [14], have become the predominant variants in many countries. The mutational patterns of these SARS-CoV-2 variants mainly vary in the S protein. As a result of the mutational patterns, the Delta and Omicron variants have been able to escape from mAb against previously reported SARS-CoV-2 variants. Furthermore, both variants exhibit a higher binding affinity between the RBD and ACE2 receptor thanks to substitutions among the Receptor Binding Motif (RBM) located in the RBD [10,15]. Indeed, the emergence of new SARS-CoV-2 variants with different mutational patterns in the S protein has triggered the development of novel effective neutralizing antibodies.

The use of Variable New Antigen Receptors (VNARs) constitutes a novel antibody technology that has gained attention for the development of new therapeutic strategies against SARS-CoV-2 [16,17]. Their low molecular weight (12 kDa) and unique structures, which are only formed by heavy chains, make VNARs attractive molecules to engage epitopes that are not available to conventional immunoglobulin antibodies [18,19]. However, the neutralizing potency of VNARs against the VOCs Delta or Omicron has not been explored. Moreover, VNAR antibodies possess four protruding paratopes that differentiate them from classical antibodies and single-domain camelid antibodies. The VNAR architecture consists of two complementarity-determining regions (CDRs) and two additional hypervariable loops (HV2 and HV4), which are pre-disposed to access and bind epitopes [20,21]. Furthermore, their stability and formulation conditions and amenability to CDR paratope reformatting make VNARs effective therapeutic molecules against the current and emerging SARS-CoV-2 variants.

In this study, we identify and characterize a new VNAR that was isolated from a synthetic phage library of VNAR domains [22]. The VNAR SP240 was found to be a potent neutralizer of the Delta and Omicron variants of authentic SARS-CoV-2 virus. The in silico analysis underscored the high affinity of SP240 to the spike protein, and, in addition, displayed the possible mechanisms of neutralization of SP240 and its likely reactivity with a broad spectrum of SARS-CoV-2 variants.

## 2. Results

### 2.1. Phage Display and VNAR Anti-Spike Purification

After four rounds of biopanning using plasma with neutralizing IgG from patients recently infected with SARS-CoV-2, a total of 72 clones were selected and sent for sequencing. We identified five VNAR with novel CDR3 sequences from the VS0 library. One of the identified sequences corresponds to the original sequence used for the VS0 library construction, which comes from a VNAR targeting TGB-β named T1. To assess the specificity of each VNAR candidate, the identified VNARs were screened against the RBD of the spike protein of SARS-CoV-2 by phage ELISA (Appendix A). After phage ELISA, four VNAR candidates were selected and obtained as recombinant proteins using a small-scale expression system of *E. coli* BL21 (DE3) (Appendix A). The reactivity of the obtained VNAR proteins was assessed by indirect ELISA against the spike protein RBD. In comparison to VNAR SP327 and T1, the VNAR SP240 confirmed the highest binding specificity for the spike RBD of SARS-CoV-2 (Figure 1).

The VNAR SP240 was expressed using an *E. coli* BL21 (DE3) strain and obtained as a soluble protein. The expression culture was supplemented with 0.1 mM of IPTG and maintained at 37 °C for 6 h, 250 rpm. The theoretical molecular weight estimated for VNAR SP240 was 15.62 kDa, including the 6-His and hemagglutinin (HA) tags. After the protein SP240 was extracted from the periplasmic space, it was subjected to IMAC purification through a Ni-NTA His-Trap column (Appendix A). In addition, a second purification step through an agarose column containing immobilized anti-HA antibodies was performed. The final purity of SP240 protein was examined in SDS-PAGE gel and the protein appeared at the expected size (Appendix A). Moreover, the amount and purity were corroborated by band densitometry. The semi-quantitative analysis indicated that the concentration of pure SP240 was 25 μg/mL. The binding ability and specificity of the obtained protein VNAR SP240 were demonstrated by indirect ELISA against its target antigen, the RBD of the spike protein (Figure 2). According to ELISA results, we corroborated that the obtained protein SP240 specifically binds to the spike’s RBD of SARS-CoV-2.

### 2.2. In Vitro SARS-CoV-2 Neutralization Assay Based on Two Cell Lines

Our in vitro model was based on the interaction blockade between the spike protein and the hACE-2 cellular receptor using the VNAR SP240. The VNAR SP240 activity was evaluated against the SARS-CoV-2 Delta and Omicron variants in the Vero E6 cell line and in the human lung cell line A549-ACE2-TMPRSS2. The tested concentration range of VNAR SP240 was from 0.1 to 10 μg/mL in two-fold dilutions. We corroborated the absence of cytopathic effects (CPE) produced by VNAR SP240 under an inverted light microscope EVOS (Figure 3). In both cell lines, no evidence of CPE or cytotoxicity triggered by the VNAR SP240 was observed even at the highest concentrations tested of SP240 10 μg/mL after 48 h of infection.

Figure 4 describes the neutralizing potency of VNAR SP240 against Delta and Omicron in the A549-ACE2-TMPRSS2 and Vero E6 cell lines (data available in Appendix A). As a negative control, we included the VNAR antibody P98Y. This VNAR P98Y has a different sequence compared to VNAR SP240 and targets the cytokine VEGF [22]. Of particular note, the low recognition activity displayed by VNAR P98Y could be caused by its binding to the spike’s RBD. The viral load of the SARS-CoV-2 Delta variant was neutralized by more than 90% starting at concentrations above 2.5 μg/mL of SP240 in A549-ACE2-TMPRSS2 (Figure 4a). Against the Omicron variant, SP240 showed an even stronger neutralizing activity, by demonstrating a complete inhibition of this variant at 2.5 μg/mL in A549-ACE2-TMPRSS2 cells (Figure 4b).

Similar results were observed in the Vero E6 cell line treated with VNAR SP240. The SARS-CoV-2 Delta variant was neutralized at 90% with 10 μg/mL of VNAR SP240 (Figure 4c). The same neutralizing activity was demonstrated against the Omicron variant of SARS-CoV-2, suppressed at 90% with 10 μg/mL of SP240 in the Vero E6 cell line (Figure 4d). However, no neutralizing effect was detected using lower concentrations of SP240 in Vero E6 cells.

Our VNAR candidate exhibited a blockade toward the SARS-CoV-2 spike protein and ACE2 interaction with a half-maximal neutralization titer (NT_50_) at nanomolar (nM) concentrations (Figure 5). The best NT_50_ values were obtained using the A549-ACE2-TMPRSS2 cell line. In the A549-ACE2-TMPRSS2 cell line, the NT_50_ of SP240 was 0.8818 μg/mL (58.01 nM) against the SARS-CoV-2 Delta variant (Figure 5a). Meanwhile, SP240 exhibited an NT_50_ of 0.1833 μg/mL (12.06 nM) in A549-ACE2-TMPRSS2 cells infected with Omicron SARS-CoV-2 (Figure 5b). To estimate the NT_50_ for the assay in Vero E6 cells, it was necessary to extrapolate concentration values beyond the tested antibody concentrations to expand the inhibition-response curve to fit the statistic metrics needed for NT_50_ determination. VNAR SP240 showed less potency in the Vero E6 cells. The NT_50_ against the SARS-CoV-2 Delta variant was 6.737 μg/mL (442.4 nM; Figure 5c), and the NT_50_ against the Omicron SARS-CoV-2 variant was 4.622 μg/mL (304 nM) in Vero E6 cells (Figure 5d). Nonetheless, in Vero E6 cells, SP240 neutralized the SARS-CoV-2 infection by 90% compared to untreated cells.

### 2.3. In Silico Analysis of Anti-Spike Antibodies: Protein–Protein Affinity and Surface Interaction Predictions

The in silico modeling revealed that the CDR3 loop from the VNAR SP240 plays a predominant role in binding to the RBD from the spike protein. The VNAR SP240 in complex with the RBD of the variants 2019-nCoV and Omicron displayed a CDR3 loop in close contact with the ACE2 binding interface (Figure 6a,b). The structural analysis suggests that the neutralization mechanism by the VNAR SP240 consists of the direct blocking of the binding surface to the ACE2 cellular receptor. Furthermore, Figure 3a,b show an atypical binding mode for VNARs to its antigen, as not only the CDR loops take part in binding the SARS-CoV-2 RBD, but also the HV2 and extensive framework residues. Figure 6c displays a surface view of the Omicron’s spike protein. The binding sites of VNAR SP240 are shown in green color. According to the position of the SP240 binding sites in the RBD, the VNAR SP240 can reach its epitopes regardless of whether the RBD protein displays its “up” or “down” conformation. Additionally, we performed a sequence alignment for the RBDs of the 2019-nCoV, Delta, and Omicron variants (Figure 6d). The alignment shows that SP240 can bind to the less conserved sites among SARS-CoV-2 variants and interacts with several mutated residues found within the receptor-binding interface of the Omicron variant. This demonstrates that the mutations do not substantially affect the SP240 binding ability, demonstrating its likely reactivity against a broad spectrum of variants.

To predict the possible binding sites of SP240 in the SARS-CoV-2 RBD and to understand the possible neutralization mechanisms we analyzed the interaction between SP240 and the RBD of SARS-CoV-2. A protein complex consisting of a recurrent neutralizing antibody VH3-53/VH3-66 (PDB ID: 7E7Y) coupled with the RBD of SARS-CoV-2 was employed as a positive binding control (Figure 7a,b). We then quantified the interaction energy (REU) to the RBD of SARS-CoV-2 for the positive control and the VNAR SP240, and a comparative analysis of the REU scores for each model is described in Table 1. The interaction score for the positive control model VH3-53/VH3-66 was −26.50 REU.

The SP240-RBD (2019-nCoV) complex scored a binding energy value of −41.97 REU. The CDR3 loop of VNAR SP240 was identified as one of the major contributors to the binding energy. Moreover, it was corroborated that the HVR2 loop plays a predominant role in binding to RBD. The interaction model indicates the HVR2 probably interacts with motifs in the spike protein needed for the ACE2 union and displayed an interaction score comparable to the estimated value for the CDR3 loop (Figure 7c,d).

A similar interaction model was observed between VNAR SP240 with Omicron’s RBD, as the SP240 uses the same peptide of the CDR3 loop to interact with the RBD (Figure 7e,f). When the VNAR SP240 is bound to the RBD from the Omicron variant, it displayed an interaction score of −41.09, and it is mainly supported by the CDR3 loop. Furthermore, an atypical collaboration to antigen binding from amino acids is observed in framework region 1 of VNAR SP240; VNARs predominantly bind to antigens using only the CDR3 loop and have poor use of framework region 1 to cooperate with antigen binding.

## 3. Discussion

During the COVID-19 pandemic, significant improvements in drug discovery were achieved. Of particular note, the mAb were the molecules preferred during the pandemic over the newly released therapeutic proteins of small size (VHH or VNAR antibodies). Some limitations that VNARs have faced are their short-serum half-life and their unknown volume required to be properly distributed systematically. Nevertheless, their simple single heavy chain structure of VNAR domains lends itself to multiple reformatting options that enable them to tackle these limitations and tailor them according to their clinical needs. Additionally, their small size and structural stability, high affinity, and specificity for their target reinforce their potential as drug candidates.

It has previously been reported that the Delta variant may show slight resistance to convalescent and vaccine-generated sera from BNT 162b2 (Pfizer/BioNTech) and ChAdOx (Oxford/AstraZeneca) [11,12]. In addition, some authorized antibody-based techniques such as the use of plasma recovered from patients have shown decreased efficiency against this novel variant [23]. The resistance shown by the SARS-CoV-2 VOCs, such as Delta and Omicron, may result in a limited therapeutic arsenal and a subsequent reduction in transmission control of these variants. The VNAR SP240 is a low molecular weight protein that promises to effectively neutralize SARS-CoV-2 VOCs. Therefore, it may be considered a novel therapeutic tool to be added to the list of therapeutic options against COVID-19.

It has been previously described that the RBD portion of the SARS-CoV-2 S protein binds with high affinity to the human ACE2 receptor while being the immunodominant target for antibodies in SARS-CoV-2 patients [4,5]. Plasma antibodies induced after infection or vaccination are highly specific and valuable tools. These antibodies are useful for diagnosing infection and evaluating antibody responses. In this study, we have shown that plasma antibodies that were induced against virus-neutralizing epitopes provided improved primary antibody selection. VNAR antibodies bound to the RBD antigen were selectively displaced by the secondary antibodies in SARS-CoV-2 positive plasma. The principle behind this selection method was antibody competition, in which one antibody displaces another antibody that recognizes the same epitope or neighboring site [24]. Several systems that follow the competitive antibody format have been described for the detection of antibodies against viral proteins in plasma by employing monoclonal antibodies (MAb) [25]. We established the flexibility of antibody competition to VNAR antibody elution during the selection rounds of the phage display. VNAR elution relies on the ability of plasma antibodies to compete with specific VNARs to bind to the RBD antigen [26]. Antibody competition not only depended on the binding affinity to the antigen but also on the concentration of antibodies in plasma. Saturation caused by high antibody titers is necessary for the plasma antibodies to compete with other high-affinity antibodies for the binding site [24]. High antibody titers in plasma are indicative of high antibody affinity and require high plasma dilutions. The plasma used in this study was tested by a Median Tissue Culture Infectious Dose (TCID_50_) assay to determine its antibody titer and neutralizing potency, and the plasma dilutions that were considered optimal for our purposes were 1:5000–1:20,000. Taking all this into account, we can ensure that the plasma antibody competed with a VNAR to bind to the RBD antigen in an epitope at the ACE2 binding interface. Additionally, a VNAR aimed at VEGF showed mild recognition activity against the tested SARS-CoV-2 variants. This supports the concept that structural motifs distributed in the RBD of SARS-CoV-2 are similar to structural motifs in interleukins and cytokines [27].

We isolated SP240 from a synthetic phage library using a plasma antibody. The identified VNAR SP240 showed a great ability to neutralize authentic virus SARS-CoV-2 in vitro. The Delta and Omicron variants were neutralized by more than 90% using concentrations of about 100 nM of SP240. VNAR SP240 showed a neutralization efficiency against SARS-CoV-2 variants comparable to other heavy chain antibodies. Humanized VHH n3130 and n3088 were able to neutralize SARS-CoV-2 pseudovirus (2019-nCoV) with NT_50_ values of 4 μg/mL and 2.5 μg/mL, respectively [28]. Similarly to the VNAR SP240, the VHH n3130 and 3088 seemed to be potent neutralizing antibodies against authentic SARS-CoV-2 variants. Other VNAR antibodies retrieved from VNAR libraries were able to neutralize the authentic SARS-CoV-2 Wuhan strain [16,17]. In contrast to the mentioned antibodies, the VNAR SP240 was tested against two authentic variants of SARS-CoV-2, Delta and Omicron, showing a strong neutralizing ability using nM concentration.

To predict the possible binding sites of VNAR SP240 to the RBD of SARS-CoV-2, the in silico models of the VNAR SP240 and the RBD were generated for protein–protein interaction analysis. The analysis highlighted the ability and plasticity of SP240 for binding to the SARS-CoV-2 variants. SP240 binds to close epitopes within the SARS-CoV-2 RBD, and its mode of interaction is completely different between variants. It was found that SP240 binds to Delta RBD via HV2 and relies on hydrophobic interactions instead of using the canonical CDR3 loop. Four residues of HV2 are involved in hydrophobic interactions between the SP240 and RBD. Trp46, Glu47, Ser48, Ile49, and Thr50 of SP240 form a hydrophobic patch with Phe490, Pro492, Leu493, and Gln494 of the RBD (Figure 7c,d). This backbone of hydrophobic interactions accounts for much of the affinity of the interaction. Other reported VNARs have been found to establish the same hydrophobic contacts in the same regions as that of the SARS-CoV-2 RBD [17]. According to the modeling results, the VNAR SP240 binds to the Omicron RBD in sites within the ACE2 interface by only using CDR3, which is mainly supported by polar interactions. Interestingly, additional framework regions of VNAR SP240 also participate in the interaction with the Omicron RBD. A comparison of the RBD sequences of the SARS-CoV-2 variants revealed that VNAR SP240 might be interacting with less-conserved sites. The Delta variant harbors two substitutions in the RBD: L452R and T478K [15]. Both substitutions stabilize and reshape the RBM loop (473–490), which contributes to the high infectivity of the Delta variant [15]. The in silico analysis suggested that SP240 binding is independent of both Delta substitutions. The Omicron variant has 10 substitutions within the RBD loop and according to in silico modeling, CDR3 of SP240 bound to four of these mutations. We have therefore shown that the VNAR SP240 is able to fit into a broad spectrum of SARS-CoV2 mutations and highlights its potential as an ideal molecule to develop antibody-based drugs against future variants.

## 4. Materials and Methods

### 4.1. Identification of SARS-CoV-2 VNARs

The VNARs targeting the SARS-CoV-2 RBD were isolated by phage display from the synthetic libraries VS0 and VS1, which were constructed by Cabanillas-Bernal et al. [22]. The target antigen was a recombinant RBD protein (expressed in *E. coli*) procured from IBT (Instituto de Biotecnologia, Universidad Nacional Autonoma de México). The phage display was conducted as previously described by Barbas et al. [29] with modifications. Briefly, about 250 ng/well of antigen RBD were immobilized in a 96-well plate overnight. Four biopanning rounds were conducted, and the number of washing steps with 1× PBS-0.05% Tween (PBST) was duplicated in each round to improve selection. Positive plasma from patients who had recently recovered from infection with SARS-CoV-2 was used for phage elution (1:5000–1:20,000 diluted plasma). The purpose of using this plasma was to compete with VNAR antibodies, displace the VNARs bound to the same antigen-binding sites, and elute phages carrying VNARs with high neutralizing potential. Random clones were selected (20–40) from panning rounds 3 and 4. Clone diversity was evaluated by sequencing the PCR product of primers Ompseq and gback [22]. The binding capacity of the clones bearing complete VNAR sequences was evaluated by phage-ELISA. The pComb3X plasmid containing sequences of interest was isolated by alkaline lysis from clones bearing VNARs with affinities to the RBD protein.

### 4.2. Production of Recombinant VNAR Antibodies

The isolated pComb3x plasmid coding to the VNAR candidates was directly transformed into electrocompetent *E. coli* BL21 (DE3) cells for further characterization. A primary culture was prepared by inoculating a single colony of transformed *E. coli* in 10 mL of SB medium, supplementing it with carbenicillin (100 μg/mL), incubating overnight at 37 °C, and shaking at 5 g. The culture was used to re-inoculate 300 mL of SB medium, which was then incubated at 37 °C until it reached an OD_600_ = 0.6–0.75. VNAR protein induction was carried out for 24 h at 37 °C with 0.1, 0.5, and 1 mM of isopropyl β-D-1-thiogalactopyranoside (IPTG). Samples from the induction cultures were taken at 3, 6, 12, and 24 h. The samples were centrifuged at 4500× *g* for 20 min, and the resulting pellet was used for periplasmic protein extraction by sonication. The lysed cells were centrifuged for 20 min at 4 °C (4500× *g*), and the supernatant was carefully collected. To further clarify the protein solution, the supernatant was passed through a 0.22-μm filter before purification by immobilized metal affinity chromatography (IMAC) via the poly-histidine tag of the VNAR protein. Protein purification was conducted in an AKTA-FPLC system using a His Trap FF 5-mL column (GE Healthcare, Chicago, IL, USA). The sample was recirculated through the column twice. Two 15-mL washes with 40 mM imidazole were performed. The protein was eluted with 200 mM of imidazole in two 5-mL fractions. The collected fractions were dialyzed in 1× PBS (SnakeSkin™ Dialysis Tubing, Thermo Fisher Scientific, Waltham, MA, USA) and assessed by 20% Tricine-SDS-PAGE gel electrophoresis as described by Schägger and von Jagow [30].

To improve protein quality, immunoaffinity purification was performed via the VNAR HA tag using Pierce^TM^ agarose (Thermo Fisher Scientific). The protein was passed through the agarose column three times. The column was then washed with 20 mL of 1× TBS followed by 25 mL of 1× TBS–0.25% Tween20 (TBST). The TBST was removed by one additional 25-mL washing step of 1× TBS. The protein was eluted from the column with 3M sodium thiocyanate–1× TBS (NaSCN-T) in 5-mL fractions. The purity was assessed by 20% Tricine-SDS-PAGE electrophoresis. Fractions containing the VNAR proteins were dialyzed in 1× TBS and quantified using a Micro-BCA protein assay kit (Thermo Fisher Scientific). VNAR bands that were revealed by SDS-PAGE electrophoresis were quantified by densitometry using ImageJ/FIJI 1.52i (NIH).

### 4.3. Antigen Recognition by ELISA

The 96-well plate was coated with 250 ng of SARS-CoV-2 RBD protein and incubated at 4 °C overnight. The coated wells were blocked with 150 μL of 5% skim milk and incubated for 2.5 h at 37 °C. A total of five washing steps with 200 μL PBS–0.5% Tween 20 were performed. A total of 50 μL of VNAR solution was added to the designated wells and incubated for 2 h at 37 °C. The solution was removed, and the PBST washings were repeated. Binding was detected using a 1:3000 dilution of anti-HA antibody coupled to peroxidase (-HRP) after incubation for 1 h at 37 °C. The five washing with PBST were repeated, and TMB substrate was added to each well. The peroxidase-substrate reaction was stopped after 15 min by adding 10% HCl, and the OD_450_ (nm) was recorded with an EPOCH reader (BioTek, Winooski, VT, USA).

### 4.4. Cell Culture

Vero E6 (VE6) and A549-ACE2-TMPRSS2 cells overexpressing human angiotensin-converting enzyme 2 (ACE2 and TMPRSS2) were used. The Vero E6 cells were purchased from ATCC (no. CRL-1586), and the A549-ACE2-TMPRSS2 cells were purchased from InvivoGen (Cat. A439-hace2tpsa). The Vero E6 cell line was maintained in DMEM medium supplemented with 10% fetal bovine serum (FBS) and 1% antibiotic/antimycotic, whereas A549-ACE2-TMPRSS2 cells were grown and kept in DMEM + 10% FBS + antibiotics (i.e., 100 μg/mL normocin, 0.5 μg/mL puromycin, and 300 μg/mL hygromycin). Both lines were maintained at 37 °C and 5% CO_2_.

### 4.5. SARS-CoV-2 Neutralization Assays

All experiments with live SARS-CoV-2 virus were conducted in a certified BSL3 laboratory, following experimental procedures as described by Friesche et al. (2022) [31] with minor modifications: Briefly, A549-AT cells/well or VeroE6 cells/well were seeded in 100 μL of DMEM media supplemented with 2% FBS (DMEM2) in a 96-well TC-treated plate 24 h prior to assay. For testing, two-fold serial dilutions of VNAR (10 μg/mL–0.1 μg/mL) were prepared in DMEM2; and pre-incubated with a TCID_50_ of 0.02 pfu/cell SARS-CoV-2 for both Delta (B.1.167.2) and Omicron (B.1.1.529) variants. The mixture was then incubated for 2 h at 37 °C prior to the assay. After incubation, the cell plate was retrieved, and the medium was removed. Then, 100 µL of VNAR/SARS-CoV-2 mixture was added to each well. Cell infection was performed at 37 °C with 5% CO_2_ for 72 h or 48 h for A549-AT cells or Vero E6 cells, respectively.

For developing, the method proposed by Franciscary Pi-Estopiñan et al. (2022) [32] was performed with modifications: After incubation, the media was removed, and the wells were washed twice with PBS (-Ca, -Mg)–0.05% Tween20 (PBST). The cells were fixed with 100 μL of 4% PFA (PFA: Sigma-Aldrich (Toluca, Mexico) #MKCJ2721) in PBS (-Ca, -Mg) for 20 min at 4 °C. After the PFA was removed, the wells were blocked with 3% BSA–PBS–0.05%Tween20 (BSA-PBST) and incubated for 30 min at room temperature. Primary antibody anti-nucleocapsid SARS-CoV-2 mouse monoclonal Ab (1:1000; SinoBiological Cat No. 40143-MM05) was prepared in blocking buffer and incubated for 1 h at room temperature. The plate was washed five times with PBST; then the secondary antibody goat anti-mouse IgG conjugated to peroxidase (1:2000; Jackson Cat #115–035-003) was prepared in a blocking buffer and added to the wells. The plate was then washed again and 50 µL of 3,3′,5,5′-Tetramethylbenzidine (TMB) Liquid Substrate System for ELISA (Sigma, Cat #T0440) was added to the wells. The reaction was stopped with 2.5 N HCl, and the plate was read at 450 nm. The data were processed in Excel v. 16.58 (Microsoft, Redmond, WA, USA). Inhibition-response curves and NT_50_ values were generated in Prism v. 8.4.0 (GraphPad Software, San Diego, CA, USA).

### 4.6. Homology Modeling, Molecular Docking, Protein-Protein Affinity, and Interaction Surface Predictions

The VNAR anti-spike sequences were subjected to BLAST-P alignment to select three VNAR scaffolds with identities higher than 50%. The template PDB files that were downloaded from the Protein Data Bank (PDB) were 4HGK, 2I26, and 2I24. The VNAR templates were used to predict the three-dimensional structures of the new anti-spike VNARs by homology modeling using MODELLER v. 9.16 [33,34]. The 3D VNAR structures were refined by simulated annealing (SA) calculations with Nanoscale Molecular Dynamics (NAMD) [35], followed by its respective analysis in Visual Molecular Dynamics (VMD) [36] and PyMOL v1.7.4.4 Edu Enhanced for MacOS.

The possible binding sites of SP240 to the SARS-CoV-2 (2019-nCoV, Delta, and Omicron variants) S protein were predicted by applying a protein–protein docking protocol using the ClusPro [37] web tool and visualized using PyMOL v. 1.7.4.4. The resulting complexes were filtered according to their binding affinity for further characterization. To predict which peptides were involved in the protein–protein interaction, we used the web tool Peptiderive from the ROSIE server with the default settings. Peptiderive scored the protein–protein interactions according to Rosetta Energy Units (REU) [38]. To obtain a reference value for a good REU score of a neutralizing SARS-CoV-2 antibody, we used a crystal structure of the SARS-CoV-2 spike RBD in complex with VH3-53-VH3-66 (BD-623) PDB: 7E7Y, resolved at 2.41 Å.

## Figures and Tables

**Figure 1 ijms-23-12267-f001:**
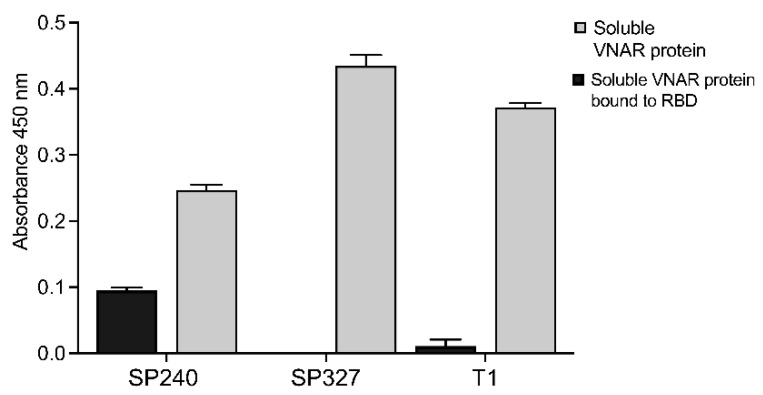
Assessment of VNAR SARS-CoV-2 binders as recombinant proteins and their recognition ability. The VNAR SARS-CoV-2 binders were obtained as soluble proteins at a small scale and their binding ability to the RBD from spike protein was evaluated by ELISA. Direct ELISA was performed to estimate the VNAR soluble protein (grey bars) obtained from an *E. coli* BL21 (DE3) expression system. The highest protein product was observed for VNAR SP327 and T1, and the lowest for VNAR SP240. However, the highest binding capacity to the RBD from the spike protein was shown by the VNAR SP240 in an indirect ELISA (black bars). We infer that other proteins might have contributed to the signal displayed for soluble proteins VNAR SP327 and T1. Indeed, the best relationship between expressed protein and target recognition was shown by VNAR SP240.

**Figure 2 ijms-23-12267-f002:**
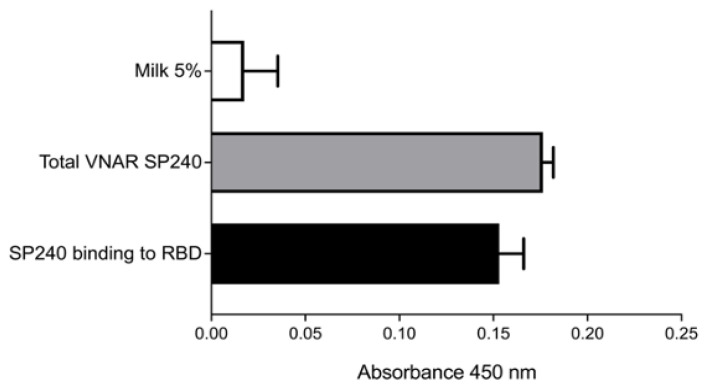
Reactivity of VNAR protein SP240 as a SARS-CoV-2 binder. The purified protein VNAR SP240 was evaluated by ELISA. The amount of total protein of SP240 and binding specificity were interpreted in terms of absorbance at 450 nm. The light grey bar corresponds to the signal displayed by total protein VNAR SP240 obtained after the purification steps. The binding reactivity of the protein VNAR SP240 to the antigen RBD of spike protein was corroborated by indirect ELISA. The black bar corresponds to the signal displayed after the binding of SP240 to RBD of the spike protein. Skim milk 5% was used as a negative control during the assay.

**Figure 3 ijms-23-12267-f003:**
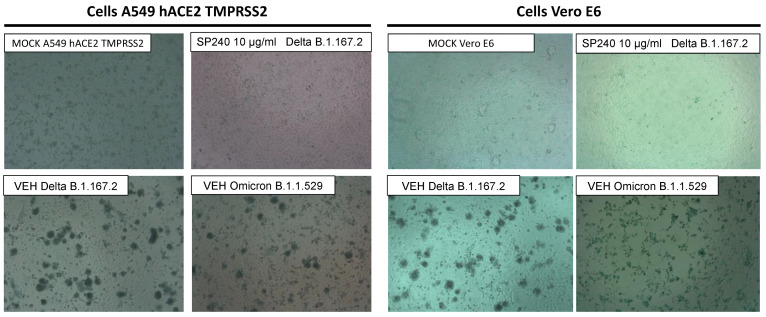
Prevention of Cytopathic Effects (CPE) in cells A549-hACE2-TMPRSS2 and Vero E6. The presence of CPE was monitored under an inverted light microscope EVOS. Displayed pictures indicate the cell’s viability after 48 h of the assay. In vehicle cells (VEH), no treatment with VNAR SP240 was reproduced. The CPE triggered by the infection of SARS-CoV-2 variants Delta (B.1.167.2) and Omicron (B.1.1.529) are shown in VEH pictures. Mock cells are not treated cells or without infection. Cells treated with 10 μg/mL of VNAR SP240 were compared to the vehicle and mock cells to exclude the presence of CPE.

**Figure 4 ijms-23-12267-f004:**
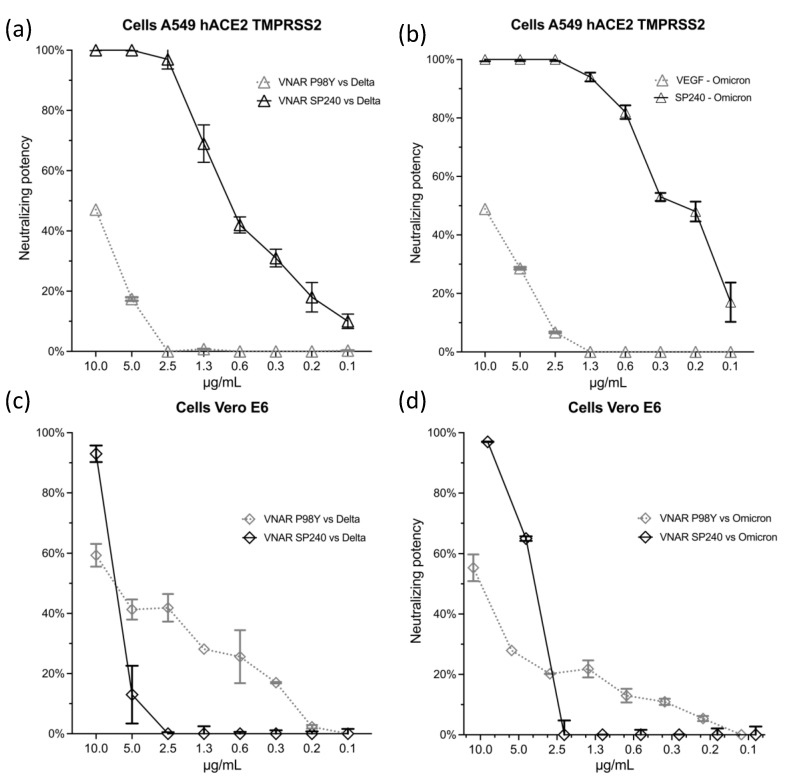
Neutralizing potency of Variable New Antigen Receptor (VNAR) SP240 against SARS-CoV-2 in A459-ACE2-TMPRSS2 and Vero E6 cell lines. The bars indicate the inhibition achieved at each tested concentration of SP240. (**a**) Delta variant neutralization by SP240 in A549-ACE2-TMPRSS2 cells. SP240 completely inhibited infection by the Delta variant from 5 μg/mL. The VNAR SP240 achieved the inhibition of Omicron infection (100%) at 2.5 μg/mL in A549-ACE2-TMPRSS2 cells (**b**). In contrast, the SP240 showed less potency against Delta (**c**) and Omicron (**d**) in the Vero E6 cell line, as a neutralizing effect was only observed with 10 μg/mL of SP240.

**Figure 5 ijms-23-12267-f005:**
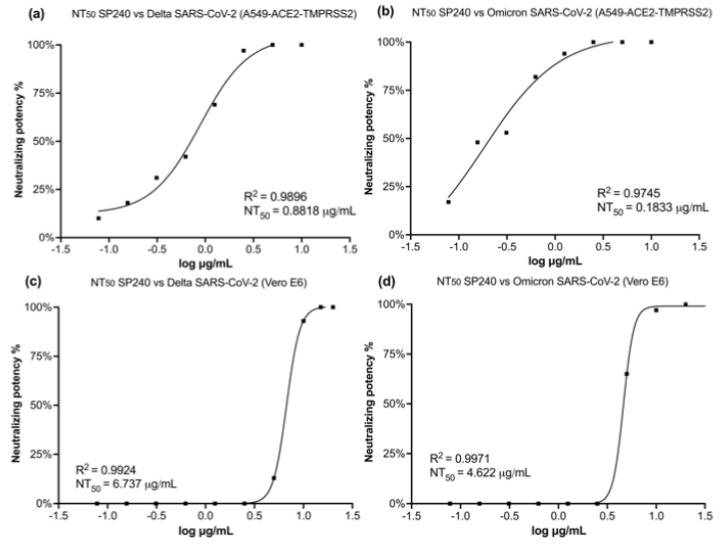
Neutralizing efficiency of Variable New Antigen Receptor (VNAR) SP240 against the SARS-CoV-2 Delta and Omicron variants. Half maximal inhibitory concentration for SARS-CoV-2 neutralization mediated by SP240 in A549-ACE2-TMPRSS2 (**a**,**b**) and Vero E6 (**c**,**d**) cells, respectively. SP240 was found to be more efficient in the A549-ACE2-TMPRSS2 cell line compared to the Vero E6 cell line.

**Figure 6 ijms-23-12267-f006:**
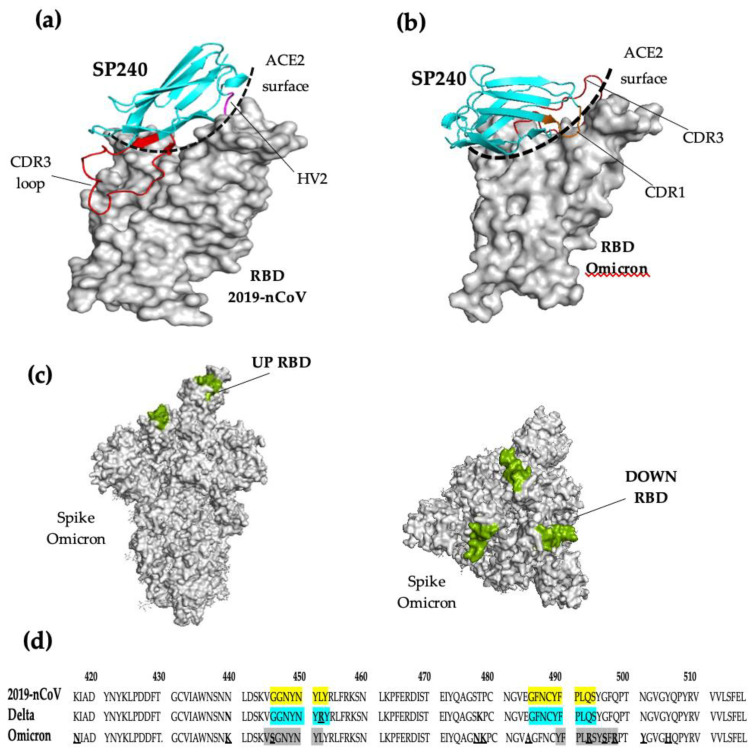
Structural analysis of Variable New Antigen Receptor (VNAR) SP240 coupled to SARS-CoV-2 variants. (**a**) VNAR SP240 coupled to the RBD of SARS-CoV-2 (2019-nCoV) and bound to the RBD of Omicron SARS-CoV-2 (**b**). In both interaction models, the CDR3 of the VNAR SP240 loop is shown in red, and the ACE2 binding interface in RBD is marked with dashes. (**c**) Surface view of the Omicron’s spike protein displaying the RBD in the “up” and “down” conformation. The binding sites of VNAR SP240 are indicated in green color. (**d**) Sequence alignments of the RBD from three SARS-CoV-2 variants. The mutated amino acids within the SARS-CoV-2 variants are underlined. The amino acids in the RBD involved in the interaction with VNAR SP240 are shown with colors: 2019-nCoV (yellow), Delta (blue), and Omicron (gray).

**Figure 7 ijms-23-12267-f007:**
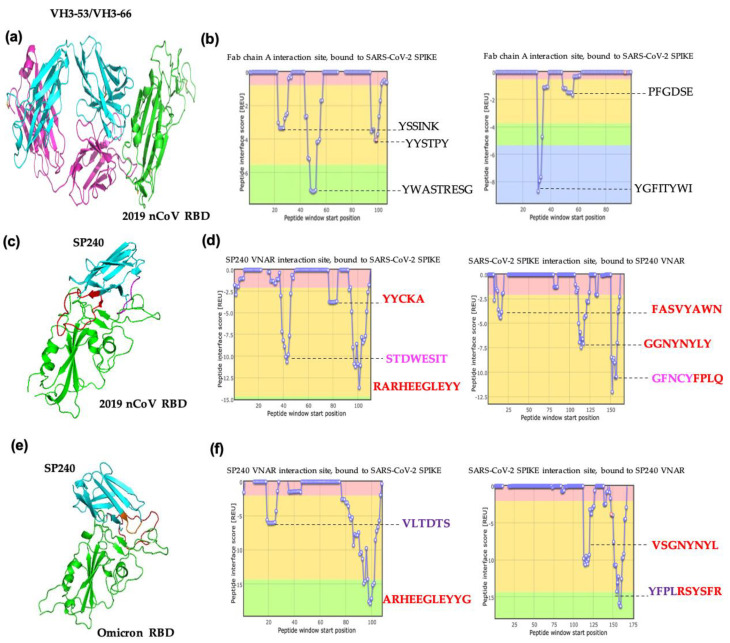
Protein–protein interaction analysis using the Peptiderive web tool. (**a**) Graphic representation of the VH3-53/VH3-66 antibody (cyan and pink) coupled to the RBD (green) of SARS-CoV-2 (2019-nCoV). (**b**) Energy plot for the interaction between the positive control VH3-53/VH3-66 and the RBD of spike protein from SARS-CoV-2 (2019-nCoV). (**c**) VNAR SP240 (cyan) coupled to spike RBD variant 2019-nCoV (green). The CDR3 loop is indicated in red color, and HV2 is shown in pink. (**d**) Energy plot for peptides in VNAR SP240 that bind the RBD of 2019-nCoV. (**e**) VNAR SP240 (cyan) coupled to the Omicron’s RBD (green). The CDR3 loop is shown in red, and the framework section involved in the interaction is indicated in orange. (**f**) Energy plots for VNAR SP240 peptides bound to the Omicron’s RBD. The CDR3 amino acids are indicated in red, HVR2 amino acids in pink, and residues from the VNAR framework in orange. VNAR SP240 sequences interacting with RBD residues are matched with colors.

**Table 1 ijms-23-12267-t001:** Interaction values of the RBD-VNAR interaction. The amino acids in the CDR3 loop that participate in the interaction are shown in bold. Mutated amino acids in the spike (S) protein are underlined.

Antibody	SARS-CoV-2 Variant	VNAR SP240 Interaction Site Bound to RBD	RBD Sites Interacting with VNAR SP240	Position in RBD	Total Score (REU)
Positive control	2019 nCoV	-	-	-	−26.50
SP240	2019 nCoV	**YYCKA** **RARHEEGLEYY** STDWESIT	FASVYAWN GGNYNYLY GFNCYFPLQS	347–354 446–453 485–494	−41.97
SP240	Omicron	VLTDTS **AREEGLEYYG**	YFPL VSGNYNYL RSYSFR	488–491 445–452 492–498	−41.06

## Data Availability

The template PDB files that were downloaded from the Protein Data Bank (PDB: https://www.rcsb.org/ accessed on 12 December 2021) were 4HGK, 2I26, and 2I24.

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
