# Peer review of "Neutralizing Ability of a Single Domain VNAR Antibody: In Vitro Neutralization of SARS-CoV-2 Variants of Concern"

_ijms, 2022, doi:10.3390/ijms232012267_

Round 1

Reviewer 1 Report

In this article, the authors test neutralizing ability of a single domain VNAR antibody for SARS-CoV-2 variants and predicted the binding epitopes in silico. Overall, the study is new for targeting Delta and Omicron variants and gives a strategy to design potential treatment for COVID-19 variants. Some revisions are needed.

Given the error bars could not be seen in Figure 1, listing the data in a supplementary table.

The data of Figure 2c listed in context and figure was not consistent, correct it.

Adjust the peptide sequence in Figure 3a to fit the figure.

In Table 1 (Omicron), change the order of spike protein position.

Line 9, (SARS-CoV-2) is abbreviated after the full name.

Line 86, E. coli should be italic.

Line 87, temperature unit should be consistent with the one in Method.

Line 309, 1X PBS >1× PBS, change all other places.

Line 326, is that 5 g?

What are the limitations of this kinds of antibodies? This should be discussed and compared with other current VNAR antibodies.

Author Response

Given the error bars could not be seen in Figure 1, listing the data in a supplementary table. Answer: As suggested by revisor 1, a Supplementary Table (Table S1) listing all the data obtained from the neutralization test. Absorbance values (492 nm), SD, neutralization potency in %, and the number of replicates for each concentration are listed on the displayed table.

The data of Figure 2c listed in context and figure was not consistent, correct it. Answer: Done. NT50 value for SP240 against SARS-CoV-2 Delta variant in Vero E6 cell line is 6.737 mg/ml, original figure number was changed to figure 5 in the revised version

Adjust the peptide sequence in Figure 3a to fit the figure. Answer: Done. All sequences described were adjusted in size to fit the figure dimensions. Original figure number was changed to figure 7 in the revised version

In Table 1 (Omicron), change the order of spike protein position. Answer: Done. In the column “Position in RBD” from Table 1, the described positions are listed in ascending order.

Line 9, (SARS-CoV-2) is abbreviated after the full name. Answer: Done

Line 86, E. coli should be italic. Answer: Done

Line 87, temperature unit should be consistent with the one in Method. Answer: Done

Line 309, 1X PBS >1× PBS, change all other places. Answer: Done

Line 326, is that 5 g? Answer: Done

What are the limitations of this kinds of antibodies? This should be discussed and compared with other current VNAR antibodies. Answer: As suggested by Revisor, we described some limitations of the VNAR antibodies compared to the currently used mAb treatments (line 261). Moreover, as we mentioned in the discussion section, the VNAR SP240 is the first VNAR antibody tested against active SARS-CoV-2 virus variants Delta and Omicron, that shows a neutralizing ability above 90% at low concentrations of antibody (line 318)

Reviewer 2 Report

The authors present a VNAR domain isolated from a synthetic phage library by panning towards RBD of SARS-CoV2 and eluting the relevant binders with positive and neutralizing SARS-CoV-2 plasma from patients who had recently recovered from infection with SARS-CoV-2. Binders have been discovered with phage ELISA and one was expressed in soluble form and tested for blocking activity in cell infection assay.  Molecular modelling, docking and delineation of interacting residues in silico has been performed for this candidate binder. As important as it is to isolate many different binders and neutralizing agents against the virus, the manuscript should be corrected in several points. In particular, some crucial experimental data should be presented and controls included. The quality of presentation, especially the clarity of figures and figure labels, should be improved. The manuscript should be re-read carefully and several sentences corrected, as at present their meaning is obscure.

My main comments are:

Monomeric state of the VNAR binder should be controlled using size exclusion chromatography in native conditions. Results demonstrating binding of the SP240 in soluble form, as well as an affinity assessment, should be provided.

Why was the SARS-CoV2 variant that was used for modelling in the control structure not used for neutralization experiments? In the Discussion section, the authors comment on the activity of SP240 on towards VOCs, but an activity comparison with this variant is not really shown.

In the neutralization assays, the prevention of cytopathic effect should be shown to demonstrate the activity of the SP240.

In the neutralization assays, a negative control (maybe VNAR with no binding activity towards RBD) and a positive control (an agent with known reactivity and neutralization activity) must be included.

Materials and methods: Please provide the details of all relevant materials used, such as antibodies RRID/ catalogue number/supplier.

Some of the sentences (please see the comments below) are written in a way that their meaning is not clear anymore. Further, colloquial expressions are often used.

In the downloaded pdf-file labelled with “original images” is a pdf version of supplementary material.

Please find a further list of comments below:

Line 22: … This study highlights the potential of VNARs to act as neutralizing antibodies against emerging SARS-CoV-2 variantsIs better than resilience.

Line 53: „Delta can escape from some mAbs”, please reword to make clear these are mAbs with neutralizing activity for certain variants of the virus

Line 59: weight

Line 72: Additionally,… and in addition, please reword

Line 74: resilience is a colloquial expression here, please reword, maybe to something like: and its likely reactivity with a broad spectrum of SARS-CoV-2 variants.

Line 78: with neutralizing IgG

Line 80: What was the frequency of the binder sequences among these 72 clones?

Line 80: Moreover, a previously reported VNAR aimed at TGF-beta was selected during the biopanning – what does this fact contribute to the manuscript? Was this VNAR binder discovered together with anti-RBD clones?

Line 83: “In a parallel ELISA, the expression levels of the VNARs on small scale were determined. Finally, the VNAR SP240 confirmed the highest specificity for the spike RBD of SARS-CoV-2.” – are these the results of the reactivity of soluble VNAR? These should be presented in the manuscript, even if in the supplementary section.

Line 93: 25 µg/ml is the concentration and not the final yield.

Line 102: I recommend using molar concentrations instead of µg/mL

Line 111: error bars are missing, also in Figure 2 (or are very small and should be presents with thicker lines).

Lines 113-118: The legend is confusing, in panel a) A459-ACE2-TMPRSS2 is stated as the cell line (and I think you meant A549), b) says A549 only, and “The Vero E6 and SP240 cell lines showed reduced potency against Delta (c) and Omicron (d)“ should be reworded as it does not mean anything.

Line 134: modest loss of potency – should be replaced with less potency

Line 138: please include the statistical analysis of the results showing the significance. The number of parallel replicates should be explicitly stated in the legend to Figures 2 and 3.

Line 146: first, the results of VNAR modeling should be described.

Table 1: The authors use interchangeably the terms RBD and Spike protein and they should correct this.

Table 1: what is Nab, neutralizing antibody?

Table 1: parts of the VNAR sequence should be presented with the labels of the corresponding region (HVR2, CDR3…). The complete sequence should also be presented, and the identified antigen-interacting sequences depicted on a structural model – with interacting RBD residues outlined (improve on Figures 3b, d and f).

Figure 3A: what is HAB, Fab?

Figure 3b: the Figure is distorted, molecular dimensions should be kept in ratio.

Figure 3d: Labels are not adjacent to the structures in the Figure

Figure 3a, c, e: something is wrong with the presentation of the sequences (line breaks), are the residues in CDR3 of C and E really different (residue H not involved in E)? The residues would be better presented with numbers and please cite the numbering scheme.

What is the orange loop in 3F and 4B?

Lines 160-161: “to interact with the Omicron RBD, which his involved in other regions, such as CDR1”, please reword, this sentence does not mean anything.

Line 196: please cite the reference for peptiderive tool.

Figure 4c. It is not clear what the labels are pointing at – please ammend

Line 216: I propose that “coupled to” is replaced by “bound to”, as “coupled” is used more often for chemical coupling

Line 227: please specify these techniques (for example: antibody-based techniques, or similar).

Line 233: a comparison of the VNAR activity with neutralizing mAbs is not really shown in the manuscript.

Line 266: I do not think the results presented here are directly comparable with the ones in reference 29 as another assay is used there, as well as other virus strains.

Lines 267-272: these lines should be modified as the authors do not show a direct comparison with any of the anti-virus antibodies listed here.

Lines 278-279: please present in your model in a figure, together with residue numbers

Line 278-279: According to which structure or which numbering scheme is the numbering here?

Line 281: please modify these sentences as the inference is based on a computational model: something like: according to the modelling results, ….. There is not really a structure or mutagenesis study presented to confirm this data.

Line 287: Here the amino acid code is one-letter, and previously 3-letter

Line 290: did you mean RBD domain?

Line 291: interacted with 4 of these amino acids.

Lines 295-300: please omit, you have already mentioned this.

Line 305: please provide the host organism and the sequence of the RBD used (or cite a reference, if possible).

Line 308: the antigen was immobilized, not fixed. What kind of plate was used?

Line 319: was the antigen used in phage ELISA RBD or Spike (as stated in supplementary Figure 1, figure legend)? Please describe the source of spike protein.

Line 311: … was used for phage elution.

Line 313: to compete with,

Line 315: was evaluated by sequencing of the PCR products

Line 316: please cite a reference where primer sequences can be found, or include them in the supplementary material

Line 336: “The sample was passed through the column three times, and two, 15-ml washes increased the imidazole concentration from 337 40 mM to 100 mM” – please reword, the sentence does not make any sense.

Lines 337-338: please re-read, way too many commas

Line 340: their purity was assessed

Lines 346-347: please list the number of additional washing steps (I think wash has no plural).

Line 357: and throughout the text: washing steps

Line 360: source of the antibody

Line 367: abbreviations for the used cell lines are defined, but not used uniformly later or previously in the manuscript

Line 380: prior to the assay

Line 404: I think that is anti-spike VNAR clones

Line 419: please include the description of the PDB structure (Fab complex)

Supplementary Figure 1: From the description in Materials and methods, it appears normalized values are shown on the y-axis.

Legend to Supplementary Figure 2b: is this SP240 after immunoaffinity purification? What precisely is the difference between E1 and E2?

Author Response

Monomeric state of the VNAR binder should be controlled using size exclusion chromatography in native conditions. Results demonstrating binding of the SP240 in soluble form, as well as an affinity assessment, should be provided. Answer: VNAR domains have not been reported in a dimeric form, that is why we didn´t consider an exclusion chromatographic step. We evaluated by an indirect ELISA the affinity and specificity of the VNAR SP240 obtained as a soluble protein (after the purification steps IMAC and Immunoaffinity) (depicted in Figure 2). The soluble protein was screened against a recombinant RBD and binding reactivity of the VNAR SP240 to the RBD was measured using a secondary IgG antibody anti-HA, aimed at the HA-tag included on the VNAR framework. We don´t have the facility to measure the real affinity (plasmon resonance).

Why was the SARS-CoV2 variant that was used for modelling in the control structure not used for neutralization experiments? In the Discussion section, the authors comment on the activity of SP240 on towards VOCs, but an activity comparison with this variant is not really shown. Answer. Since the expression of VNAR for testing (recombinant produced protein) was in very low amounts, we tested activity against circulating variants at the moment of the experiments, with the rationale that those variants were and will be the predominant worldwide.

In the neutralization assays, the prevention of cytopathic effect should be shown to demonstrate the activity of the SP240. Answer. As suggested by reviewer, photographs showing no CPE on neutralization over A549-AT and Vero E6 are provided in a new Figure 3 on the revised version

In the neutralization assays, a negative control (maybe VNAR with no binding activity towards RBD) and a positive control (an agent with known reactivity and neutralization activity) must be included. Answer. As suggested by the reviewer, an extra experiment testing anti-VEGF VNAR was performed in Vero E6 and A549-AT, this molecule has a different sequence compared to SP240 VNAR. In Figure 4 on the revised version, you may find the modified plots, including the anti-VEGF antibody (P98Y antibody). We associate the low neutralizing activity displayed by VNAR anti-VEGF to a cross-reaction with the Spike RBD of SARS-CoV-2. It has been previously reported the presence of structural motifs among the spike protein that possess structural similarities with domains of cytokines and interleukins. However, as it showed in our results the cross-reactivity of VNAR anti-VEGF to the RBD does not guarantee complete viral neutralization. Moreover, a neutralizing effect of 50% is only visible with the highest concentration tested for anti-VEGF vNAR.

Materials and methods: Please provide the details of all relevant materials used, such as antibodies RRID/ catalogue number/supplier. Answer: as suggested by the reviewer the next were included in the text of material and methods

Some of the sentences (please see the comments below) are written in a way that their meaning is not clear anymore. Further, colloquial expressions are often used. Answer. Done

In the downloaded pdf-file labelled with “original images” is a pdf version of supplementary material. Answer. This is an error of downloading or naming in the original documents, it was checked and resolved.

Please find a further list of comments below:

Answer. All comments were check and changed accordingly:

  • Line 22: …“ This study highlights the potential of VNARs to act as neutralizing antibodies against emerging SARS-CoV-2 variants“ Is better than resilience.
  • Line 53: „Delta can escape from some mAbs”, please reword to make clear these are mAbs with neutralizing activity for certain variants of the virus
  • Line 59: weight
  • Line 72: Additionally,… and in addition, please reword
  • Line 74: resilience is a colloquial expression here, please reword, maybe to something like: and its likely reactivity with a broad spectrum of SARS-CoV-2 variants.
  • Line 78: with neutralizing IgG

Line 80: What was the frequency of the binder sequences among these 72 clones? R. The observed frequency of each VNAR SARS-CoV-2 binder was just one. We attribute this to the specific selection offered by the phage elution with plasma. The normal phage display technology with Lysozyme, brings 7-12 same clones, but in this approach, we used hyperimmune serum from a patient in order to elute by competition the neutralizing VNARs.  In addition, very high plasma dilutions were used for phage elution, from 1:5000 to 1:20000. In comparison to an enzyme (trypsin), using plasma with neutralizing antibodies offers a specific and delicate selection of only those VNAR that are specifically bound to neutralizing epitopes in the Spike RBD.

Line 80: Moreover, a previously reported VNAR aimed at TGF-beta was selected during the biopanning – what does this fact contribute to the manuscript? Was this VNAR binder discovered together with anti-RBD clones? Answer. VNAR aimed at TGF-beta was selected during the phage display with the anti-RBD clones. The library used for the phage display against the RBD was previously constructed using the sequence of the VNAR anti-TGF-beta as the framework. During the library amplification, some clones bearing the library's original sequence might be amplified.

Line 83: “In a parallel ELISA, the expression levels of the VNARs on small scale were determined. Finally, the VNAR SP240 confirmed the highest specificity for the spike RBD of SARS-CoV-2.” – are these the results of the reactivity of soluble VNAR? These should be presented in the manuscript, even if in the supplementary section. Answer: yes, they are part of the kinetic of expression defined in supplementary material figure

  • Line 93: 25 µg/ml is the concentration and not the final yield. Answer: Done
  • Line 102: I recommend using molar concentrations instead of µg/mL Answer: Done
  • Line 111: error bars are missing, also in Figure 2 (or are very small and should be presents with thicker lines). Answer: check and changed

Lines 113-118: The legend is confusing, in panel a) A459-ACE2-TMPRSS2 is stated as the cell line (and I think you meant A549), b) says A549 only, and “The Vero E6 and SP240 cell lines showed reduced potency against Delta (c) and Omicron (d) “should be reworded as it does not mean anything. Answer: Done and changed

Line 134: modest loss of potency – should be replaced with less potency Answer: Done and changed

Line 138: please include the statistical analysis of the results showing the significance. The number of parallel replicates should be explicitly stated in the legend to Figures 2 and 3. Answer. Table S1 in the supplementary material is included in the revised version

Line 146: first, the results of VNAR modeling should be described.

Table 1: The authors use interchangeably the terms RBD and Spike protein and they should correct this. Answer: Done

Table 1: what is Nab, neutralizing antibody? Answer: yes it is, check and changed

Table 1: parts of the VNAR sequence should be presented with the labels of the corresponding region (HVR2, CDR3…). The complete sequence should also be presented, and the identified antigen-interacting sequences depicted on a structural model – with interacting RBD residues outlined (improve on Figures 3b, d and f). Answer. As the head table mention, we present only the CDR3 aminoacids of the VNAR that interact with the RBD protein. We did not present the complete VNAR sequence, because we are submitting a patent to protect the use of this VNAR.

Figure 3A: what is HAB, Fab? Answer: changed, new figure

Figure 3b: the Figure is distorted; molecular dimensions should be kept in ratio. Answer: it was changed from the original and corrected, new version figure is #7 in the reviewed version

Figure 3d: Labels are not adjacent to the structures in the Figure. Answer. Figure 7 in the new version changed

Figure 3a, c, e: something is wrong with the presentation of the sequences (line breaks), are the residues in CDR3 of C and E really different (residue H not involved in E)? The residues would be better presented with numbers and please cite the numbering scheme. Answer. The involved residues are different, because the VNAR do not binds to the same epitope (in silico) in variant delta and in vvariant omicron as described in the new figure 7.

What is the orange loop in 3F and 4B? Answer. is the CDR1

Lines 160-161: “to interact with the Omicron RBD, which his involved in other regions, such as CDR1”, please reword, this sentence does not mean anything. Answer. check and changed

Line 196: please cite the reference for peptiderive tool. Answer. check and changed

Figure 4c. It is not clear what the labels are pointing at – please amend Answer. check and changed

Line 216: I propose that “coupled to” is replaced by “bound to”, as “coupled” is used more often for chemical coupling Answer. check and changed

Line 227: please specify these techniques (for example: antibody-based techniques, or similar). Answer. Done

Line 233: a comparison of the VNAR activity with neutralizing mAbs is not really shown in the manuscript. Answer. as requested, the comparison is performed in the revised version and is included in the experimental and discussion parts

Line 266: I do not think the results presented here are directly comparable with the ones in reference 29 as another assay is used there, as well as other virus strains. Answer. We were trying to argue that single domain antibodies could be used for SARS-CoV-2 neutralization.

Lines 267-272: these lines should be modified as the authors do not show a direct comparison with any of the anti-virus antibodies listed here. Answer. As we mention above, we just want to shown that the use of single domain antibodies have the potential to be used as drugs against SARS-CoV-2, we are not suggesting that our VNAR is better that the other examples, because we are agree whit the reviewer, they use another assays.

Lines 278-279: please present in your model in a figure, together with residue numbers. Answer. This is described at figure 7c and 7d, we add this information at the manuscript.

Line 278-279: According to which structure or which numbering scheme is the numbering here? Answer. These are the nombers of the whole secuence of the VNAR or the RBD protein.

Line 281: please modify these sentences as the inference is based on a computational model: something like: according to the modelling results... There is not really a structure or mutagenesis study presented to confirm this data. Answer. Thanks for the comment, done.

Line 287: Here the amino acid code is one-letter, and previously 3-letter. Answer. We have seen that the mutation position, is represented with one-letter code, example: L453P, instead of Leu453Pro.

Line 290: did you mean RBD domain? Answer. check and changed

Line 291: interacted with 4 of these amino acids. Answer. check and changed

Lines 295-300: please omit, you have already mentioned this. Answer. check and changed

Line 305: please provide the host organism and the sequence of the RBD used (or cite a reference, if possible). Answer: done.

Line 308: the antigen was immobilized, not fixed. What kind of plate was used? Answer: check and modified on text, plate used was high binding surface u-bottom plate from corning

Line 319: was the antigen used in phage ELISA RBD or Spike (as stated in supplementary Figure 1, figure legend)? Please describe the source of spike protein. Answer.  It was RBD, it was modified.

Line 311: … was used for phage elution. Answer. Done, thanks

Line 313: to compete with, Answer. Done, thanks

Line 315: was evaluated by sequencing of the PCR products. Answer. Done.

Line 316: please cite a reference where primer sequences can be found, or include them in the supplementary material. Answer. As we mention, we didn´t shown the sequence because of patent issues.

Line 336: “The sample was passed through the column three times, and two, 15-ml washes increased the imidazole concentration from 337 40 mM to 100 mM” – please reword, the sentence does not make any sense. Answer. Done

Lines 337-338: please re-read, way too many commas. Answer. Done

Line 340: their purity was assessed Answer. Done

Lines 346-347: please list the number of additional washing steps (I think wash has no plural). Answer. Done

Line 357: and throughout the text: washing steps. Answer. Done

Line 360: source of the antibody. Answer. Done

Line 367: abbreviations for the used cell lines are defined, but not used uniformly later or previously in the manuscript. Answer. Done

Line 380: prior to the assay. Answer. Done

Line 404: I think that is anti-spike VNAR clones. Answer. Done

Line 419: please include the description of the PDB structure (Fab complex). Answer. Done

Supplementary Figure 1: From the description in Materials and methods, it appears normalized values are shown on the y-axis. Answer.It is correct.

Legend to Supplementary Figure 2b: is this SP240 after immunoaffinity purification? What precisely is the difference between E1 and E2? Answer. E1 and E2 correspond to different fractions in which VNAR SP240 protein was eluted. Due to the established purification program, the protein was collected in two different elution fractions. At the end of the purification, all fractions were pooled together.

Round 2

Reviewer 2 Report

Dear Authors,

I have seen you have undertaken some corrections of the manuscript, but the figures have not been changed from the first version. Although you have answered the questions in the "Authors' reply" and included the comments on novel Figures (you mention for example modification of Figure 3, modification of Figure 4, Figure 7 etc.), they are not really modified. Can it be that a wrong version of the manuscript was uploaded?

Author Response

We apologies for this mistake.

Round 3

Reviewer 2 Report

The authors have supplemented the manuscript with additional data and experimental controls, which has enhanced its value. Also, some suggested experiments that were not performed (such as the demonstration of the prevention of the cytopathic effect by VNAR SP240) were additionally explained by adding references. Others could not be performed because the equipment is not available. Neutralization assay still misses a positive control, which should be validated and described before (in the publicly available literature), so that the NT50 values of the novel inhibitor SP240 can be compared.

Some statements in the manuscript are still not that well founded. For example, the discovery of the anti-TGF-beta VNAR should be explained to the reader as well, at the moment it is not clear if it has a biological meaning or is it only a library screening artefact. The important negative control anti-VEGF VNAR shows some effect in the neutralization assay, and this is explained as its cross-reactivity with the RBD. The authors should examine its binding with RBD antigen.

Something is wrong with the order of references: e.g. reference labelled as 30 that describes the mixture of neutralizing VHHs appears in the reference list as 29.

The authors have corrected the original article for several unclarities in the expression and the text is really better now. The newly written paragraphs are in contrast not so precise (please also see comments below) and I propose a careful read-through and correction by all authors.

Please find a list of further remarks below.

Line 82: please reword:

One of the identified sequences corresponds to a previously reported VNAR aimed at TGB-beta, which was a part of the parental vector used for cloning (unless you mean something else).

Figure 1, legend in the Figure: bound to RBD

Line 99: “We don’t discard other proteins that might have contributed to the signal displayed for soluble proteins VNAR SP327 and T1”, I do not understand this sentence: did you mean exclude the possibility that other proteins might have contributed…?

Line 109: Please reword: the purity of the of SP240 protein was examined in SDS-PAGE gel and the protein appeared at the expected size

Line 111: amount of pure SP240 was 25 ug/ml: concentration and not amount

Line 123: According to the detected signals, more than 80% of the total VNAR SP240 protein is binding to the RBD of Spike protein from SARS-CoV-2.: I propose that this sentence is omitted: if you coat differently, maybe the reactivity of the SP240 will change

Line 143: compared to the vehicle and mock cells to discard the presence of CPE: to exclude the presence of CPE.

Line 149: was the cross-reactivity of the anti-VEGF VNAR with RBD examined?

Line 151: does not guarantee

Line 154: SP240 showed improved 154 neutralizing activity: showed an even stronger neutralizing activity

Figure 5. Graphs are skewed.

Line 244: Furthermore, it is observed an atypical collaboration to binding from amino acids in framework region 1 of VNAR SP240“, did you mean an atypical contribution to antigen binding by the amino acids in FR1 was observed?

Figure 7A. Fab model is still skewed.

Figure 7B. What does HAB mean?

Figure 7F. Do not use orange and red color together, they are very similar.

Line 277: “The VNAR SP240 is a low molecular weight protein that promises to achieve the complete neutralization of SARS-CoV-2 VOCs.“ With the data presented, this is a very daring statement. Even if in the in vitro assays 100% neutralization can be achieved, which is great, this must not be the case with other cells and not at all in vivo. Even with stronger inhibitory agents, mixtures have proven beneficial for neutralization. Please reword.

Line 305: “During panning we selected a previously reported VNAR aimed at TGF-beta”. Please comment if this only indicates the remaining parental vector, or does it have any biological meaning?

Line 306: “a VNAR aimed at VEGF showed mild neutralization activity against the tested SARS-CoV-2 variants. This supports the concept that structural motifs distributed in the RBD of SARS-CoV-2 are similar to structural motifs in interleukins and cytokines.” Please establish if there is any cross-reactivity of the anti-VEGF binder with RBD or spike. In this case, please use a different negative control.

Line 312: “The Delta and Omicron variants were neutralized by more than 90% using nanomolar concentrations of SP240“. Figure 4 suggests that these concentrations were about 100 nM. Please amend.

Line 313: “SP240 probed a neutralization efficiency“: showed neutralization efficiency

Line 319: “Nevertheless, the VNAR SP240 is the first VNAR antibody tested against live SARS-CoV-2 virus variants Delta and Omicron, that shows a neutralizing ability of 100% at nM potency”: as mentioned, these are high nanomolar concentrations. nM indicates concentration and not potency. As the authors cannot confirm that their VNAR is monomeric, this statement should be changed not to be misleading.

Line 322: please indicate clearly that the below statements are inferred from a model, and not an actual structure.

Line 352: immobilized and not fixed, or otherwise please specify the fixation method

Line 353: washing steps, not washes

Line 356: (1:5,000–1:20,000) please specify if this was the plasma dilution (it could be antibody titer, neutralization titer or similar)

Line 359: from panning rounds 3 and 4

Line 381: mM imidazole, mm is millimeter

Line 401: washing steps

Line 426: TCID50, 50 in subscript. Does this concentration present the half-maximal for both cell lines used? How was the TCID50 determined?

Line 474: Supplementary materials also contains Supplementary Tables

Table S1 and S2: “lectures at 492 nm“ should be readings. Please dear authors check that 492 is really the correct wavelength because TMB absorbs at 450 nm, usually.

Supplementary Figure 1, Legend:

The binding of each candidate to the RBB“ should be RBD

Supplementary Figure 2, Legend: please replace the expression “retrieved” with “isolated”.

Author Response

The authors have supplemented the manuscript with additional data and experimental controls, which has enhanced its value. Also, some suggested experiments that were not performed (such as the demonstration of the prevention of the cytopathic effect by VNAR SP240) were additionally explained by adding references. Others could not be performed because the equipment is not available. Neutralization assay still misses a positive control, which should be validated and described before (in the publicly available literature), so that the NT50 values of the novel inhibitor SP240 can be compared.

Some statements in the manuscript are still not that well founded. For example, the discovery of the anti-TGF-beta VNAR should be explained to the reader as well, at the moment it is not clear if it has a biological meaning or is it only a library screening artefact. The important negative control anti-VEGF VNAR shows some effect in the neutralization assay, and this is explained as its cross-reactivity with the RBD. The authors should examine its binding with RBD antigen.

Something is wrong with the order of references: e.g. reference labelled as 30 that describes the mixture of neutralizing VHHs appears in the reference list as 29.

The authors have corrected the original article for several unclarities in the expression and the text is really better now. The newly written paragraphs are in contrast not so precise (please also see comments below) and I propose a careful read-through and correction by all authors.

Please find a list of further remarks below.

Line 82: please reword:

One of the identified sequences corresponds to a previously reported VNAR aimed at TGB-beta, which was a part of the parental vector used for cloning (unless you mean something else). CHANGED

Figure 1, legend in the Figure: bound to RBD CHANGED

Line 99: “We don’t discard other proteins that might have contributed to the signal displayed for soluble proteins VNAR SP327 and T1”, I do not understand this sentence: did you mean exclude the possibility that other proteins might have contributed…? CHANGED

R: The sentence was reworded to “We infer that other proteins might have contributed to the signal displayed for soluble proteins VNAR SP327 and T1”

Line 109: Please reword: the purity of the of SP240 protein was examined in SDS-PAGE gel and the protein appeared at the expected size CHANGED

Line 111: amount of pure SP240 was 25 ug/ml: concentration and not amount CHANGED

Line 123: According to the detected signals, more than 80% of the total VNAR SP240 protein is binding to the RBD of Spike protein from SARS-CoV-2.: I propose that this sentence is omitted: if you coat differently, maybe the reactivity of the SP240 will changeCHANGED

Line 143: compared to the vehicle and mock cells to discard the presence of CPE: to exclude the presence of CPE. CHANGED

Line 149: was the cross-reactivity of the anti-VEGF VNAR with RBD examined? CHANGED

We decided to reword line 149: Of particular note, the low neutralizing activity displayed by VNAR P98Y could be caused by its binding to the Spike’s RBD. Nonetheless, the low neutralizing activity reached by P98Y suggests that it might be binding outside of the interaction surface to ACE2.

Line 151: does not guarantee CHANGED

Line 154: SP240 showed improved 154 neutralizing activity: showed an even stronger neutralizing activity CHANGED

Figure 5. Graphs are skewed. CHANGED

Line 244: Furthermore, it is observed an atypical collaboration to binding from amino acids in framework region 1 of VNAR SP240“, did you mean an atypical contribution to antigen binding by the amino acids in FR1 was observed? changed

 Figure 7A. Fab model is still skewed. changed

Figure 7B. What does HAB mean?changed

Figure 7F. Do not use orange and red color together, they are very similar.changed

Line 277: “The VNAR SP240 is a low molecular weight protein that promises to achieve the complete neutralization of SARS-CoV-2 VOCs.“ With the data presented, this is a very daring statement. Even if in the in vitro assays 100% neutralization can be achieved, which is great, this must not be the case with other cells and not at all in vivo. Even with stronger inhibitory agents, mixtures have proven beneficial for neutralization. Please reword. CHANGED

CHANGED TO: Line 289 The VNAR SP240 is a low molecular weight protein that promises to effectively neutralize SARS-CoV-2 VOCs.

Line 305: “During panning we selected a previously reported VNAR aimed at TGF-beta”. Please comment if this only indicates the remaining parental vector, or does it have any biological meaning? Answer: It is the remaining parental vector, we eliminate this sentence, it was causing confusion to lector.

Line 306: “a VNAR aimed at VEGF showed mild neutralization activity against the tested SARS-CoV-2 variants. This supports the concept that structural motifs distributed in the RBD of SARS-CoV-2 are similar to structural motifs in interleukins and cytokines.” Please establish if there is any cross-reactivity of the anti-VEGF binder with RBD or spike. In this case, please use a different negative control. Answer: The VNAR against VEGF recognize poorly the spike protein, however, this recognition cannot be considered as neutralization activity.

 Line 312: “The Delta and Omicron variants were neutralized by more than 90% using nanomolarconcentrations of SP240“. Figure 4 suggests that these concentrations were about 100 nM. Please amend. CHANGED 

Line 313: “SP240 probed a neutralization efficiency“: showed neutralization efficiency CHANGED

 Line 319: “Nevertheless, the VNAR SP240 is the first VNAR antibody tested against live SARS-CoV-2 virus variants Delta and Omicron, that shows a neutralizing ability of 100% at nM potency”: as mentioned, these are high nanomolar concentrations. nM indicates concentration and not potency. As the authors cannot confirm that their VNAR is monomeric, this statement should be changed not to be misleading. CHANGED

Reworded in line 329: In contrast to the mentioned antibodies, the VNAR SP240 was tested against two authentic variants of SARS-CoV-2, Delta and Omicron, showing a strong neutralizing ability using nM concentration.

Line 322: please indicate clearly that the below statements are inferred from a model, and not an actual structure. CHANGED

Reworded in line 351: To predict the possible binding sites of VNAR SP240 to the RBD of SARS-CoV-2, the in silico models of the VNAR SP240 and the RBD were generated for protein-protein interaction analysis. The analysis highlighted the ability and plasticity of SP240 for binding to the SARS-CoV-2 variants.

Line 352: immobilized and not fixed, or otherwise please specify the fixation method CHANGED

Line 353: washing steps, not washes CHANGED

Line 356: (1:5,000–1:20,000) please specify if this was the plasma dilution (it could be antibody titer, neutralization titer or similar)CHANGED

Line 359: from panning rounds 3 and 4 CHANGED

Line 381: mM imidazole, mm is millimeter CHANGED

Line 401: washing steps CHANGED

Line 426: TCID50, 50 in subscript. Does this concentration present the half-maximal for both cell lines used? yes How was the TCID50 determined? TCID50 was calculated by the Reed–Muench Method

Line 474: Supplementary materials also contains Supplementary Tables CHANGED

Table S1 and S2: “lectures at 492 nm“ should be readings. Please dear authors check that 492 is really the correct wavelength because TMB absorbs at 450 nm, usually.  CHANGED

Supplementary Figure 1, Legend: The binding of each candidate to the RBB“ should be RBD CHANGED

Supplementary Figure 2, Legend: please replace the expression “retrieved” with “isolated”. CHANGED

Round 4

Reviewer 2 Report

The authors have corrected most of the ambiguities in the text and included additional comments on the experimental results. I understand that some of the suggested improvements which would include additional experiments cannot be undertaken. I would like to ask the authors for a few corrections:

Line 244: Furthermore, it is observed an atypical collaboration to binding from amino acids in framework region 1 of VNAR SP240“, did you mean an atypical contribution to antigen binding by the amino acids in FR1 was observed? Although the authors response letter promises that the sentence is changed, it is not changed. Please correct.

Supplementary Figure 1. Please change RBB to RBD.

“Table S1. In vitro assays for neutralizing activity of VNAR SP240. Statistical analysis of data obtained after lectures at 492 nm. Table S2. In vitro assays for neutralizing activity of VNAR P98Y. Statistical analysis of data obtained after lectures at 492 nm.”

The authors have corrected the measurement wavelength to be 450 nm in the main text (line 443), but not in the legends to supplementary tables, which are also cited when listing “Supplementary material” in the main document. “Lectures” should be readings.

Supplementary material includes western blots, but they are not mentioned in the main document.

Author Response

The authors have corrected most of the ambiguities in the text and included additional comments on the experimental results. I understand that some of the suggested improvements which would include additional experiments cannot be undertaken. I would like to ask the authors for a few corrections: We sincerely appreciate all the comments of the reviewer. All of them helped us to have an improved manuscript.

Line 244: Furthermore, it is observed an atypical collaboration to binding from amino acids in framework region 1 of VNAR SP240“, did you mean an atypical contribution to antigen binding by the amino acids in FR1 was observed? Although the authors response letter promises that the sentence is changed, it is not changed. Please correct. Answer: We have rephrased the sentence to increase clarity.

Supplementary Figure 1. Please change RBB to RBD. Answer: Done

“Table S1. In vitro assays for neutralizing activity of VNAR SP240. Statistical analysis of data obtained after lectures at 492 nm. Table S2. In vitro assays for neutralizing activity of VNAR P98Y. Statistical analysis of data obtained after lectures at 492 nm.”

The authors have corrected the measurement wavelength to be 450 nm in the main text (line 443), but not in the legends to supplementary tables, which are also cited when listing “Supplementary material” in the main document. “Lectures” should be readings.  Answer: Done

Supplementary material includes western blots, but they are not mentioned in the main document. Answer: it is not a western blot. Figure S3-a it’s the PAGE-SDS of imac purification, and figure S3-b it’s a PAGE-SDS of an immunoaffinity purification. We have changed the text